# Characterization of the Volatiles and Quality of Hybrid Grouper and Their Relationship to Changes of Microbial Community During Storage at 4 °C

**DOI:** 10.3390/molecules25040818

**Published:** 2020-02-13

**Authors:** Wenbo Huang, Jing Xie

**Affiliations:** 1College of Food Science and Technology, Shanghai Ocean University, Shanghai 201306, China; wenbohuang1@163.com; 2Shanghai Engineering Research Center of Aquatic Product Processing and Preservation, Shanghai 201306, China; 3National Experimental Teaching Demonstration Center for Food Science and Engineering (Shanghai Ocean University), Shanghai 201306, China

**Keywords:** hybrid grouper, microbiota diversity, VOCs, *Pseudomonas*

## Abstract

To investigate the effects of spoilage bacteria on aquatic product quality and volatile organic compounds (VOCs) in hybrid grouper (*Epinephelus fuscoguttatus* ♀×*Epinephelus lanceolatus* ♂), the physical conditions were evaluated, the chemical changes including color, total volatile base nitrogen (TVB-N), VOCs, and free amino acids (FFAs) were determined, and biological profiles were made through microbial community (total viable counts (TVC), 16S rRNA gene amplification sequencing, and next-generation sequencing (NGS) technology on hybrid grouper, which were stored at 4 °C for 10 days. The results showed that the whiteness and TVB-N of grouper increased throughout the storage period. The contents of glycine, alanine, and total free amino acid decreased with the microbial activity towards the end of the study period. At the end of storage, the TVC reached 9.0 log10 (CFU/g). Seventy eight strains of bacteria were isolated from the hybrid grouper, most of which were shown to be *Pseudomonas spp.*, after 16S rRNA sequencing. The results of the NGS test showed that the diversity of dominant bacteria decreased with time; *Pseudomonas azotoformans* was the dominant spoilage bacteria at the end of storage. The VOCs of fish and bacteria in the grouper’s spoilage process were presented in headspace solid-phase microextraction gas chromatography-mass spectrometry (HS-SPME-GC-MS). Twenty eight compounds were identified in hybrid grouper, among which alcohol and aldehyde were used to characterize freshness, both of which were not only related to the overall flavor of the grouper, but were also affected by microbial activity. However, due to the complexity of microbial communities in aquatic products, the correlation between community changes and VOCs needs further research. This study provides insights into the correlation between VOCs and specific spoilage organisms (SSOs) through the analysis of the microbial community and VOCs.

## 1. Introduction

Aquatic products are easily perishable due to biological and chemical changes. Along with the quality deterioration caused by microbial growth, the odor and flavor of fish undergo great shifts that directly influence customers’ purchasing behavior [1]. In fish spoilage, various microorganisms produce compounds with special odors such as nitrogenous compounds, aldehydes, ketones, and esters [2,3,4]. Therefore, they are primarily used to profile the composition of microbiota, and moreover, evaluate changes in quality and their relation with the quality and composition of microbiota during fish storage.

Under particular storage conditions, the cause of off-flavors and organoleptic rejection of seafood are related to a consortium of the metabolites (chemical spoilage indices, CSIs) produced by bacteria called Specific Spoilage Organisms (SSOs) [2]. These metabolites include a range of volatile organic compounds (VOCs). The headspace solid-phase microextraction gas chromatography-mass spectrometry (HS-SPME-GC-MS) is a useful analytical method for testing for VOCs, which can be used to identify potential strategies for quality assessments and for estimates of remaining shelf life [5,6,7,8,9]. This technique has been used to analyze changes in volatile compounds in the headspace of yellowfin tuna [6], raw Atlantic salmon [7], mackerel [8], and gilthead sea bream fish [10]. It provides valuable information with which to assess the extent of seafood deterioration.

The potential spoilage microbiota in some seafood has not been investigated [11]. For example, hybrid grouper (*Epinephelus fuscoguttatus* ♀×*Epinephelus lanceolatus* ♂) has the advantages of fast growth, high collagen egg content, and good taste. Grouper is a high-quality commercial fish which is rich in amino acids and unsaturated fatty acids. It has become the new darling of grouper fish and has broad market prospects. Its nutrients and delicacies are very popular in many countries.

In this study, the composition and quality changes of hybrid grouper during storage at 4 °C for 10 days were selected as the research objects. VOCs were used to determine the degree of spoilage of hybrid grouper. The purpose of this study is to provide a more comprehensive understanding of the dominant spoilage bacteria in cold-stored groupers, and to provide key ideas for fresh-keeping methods.

## 2. Results and Discussion

### 2.1. Physico-Chemical Analysis

Visible appearance is an important factor determining the perceived freshness and market value of fish fillets [12]. Therefore, color becomes a determinant of consumer acceptance and marketability; it is always considered an important factor, reflecting product quality [13,14]. The formation of color is attributed to the oxidation of proteins and lipids, which are abundant in fish [15]. Table 1 reveals that W* increased from 41.46 ± 1.37 (day 0) to 47.45 ± 0.77 (day 10) over the period, and the value of *L** increased as well. This phenomenon was consistent with the discoloration of channel catfish during storage [16]. However, there was a fluctuation on day 6, which may be have been caused by external factors such as the size of the fish, heterogeneity, and the feeding ingredients of the batches of fish [17].

The TVB-N of grouper fillets during cold storage is shown in Table 1. The initial TVB-N of fresh fish (day 0) was 9.60 mg/100 g, followed by a mild increase over 10 days. It has been reported that the production of TVB-N in fish is caused by bacterial proliferation and bacterial decomposition of trimethylamine oxide (TMAO) and amino acid side chains [18,19,20].

### 2.2. Free Amino Acid Analysis

FAAs are responsible for the formation of aroma, which is one of the main factors in the development of taste, such as sweetness, umami, and bitterness. FFAs are also precursors of harmful biogenic amines. For example, histidine is the precursor of histamine [21]; it is very important from the perspective of toxicity and is used as a quality indicator for various fish [22,23]. Table 2 shows the variations of FAAs in grouper during storage. The most abundant FAAs were threonine, glycine, and lysine. Interestingly, the amount of total FAAs, as well as threonine, glycine, and alanine, increased within 2 days of storage and then decreased until the end of storage, agreeing with the observations of Shi [24] on silver carp and Ruiz-Capillas [25] on hake. Scholars have speculated that the increase during early storage may be due to muscle autolysis, inducing the breakdown of dipeptides caused by proteolytic enzymes [26], whereas the later decrease may be due to the proliferation of microorganisms, with which amino acids can produce biogenic amines such as tyrosine, arginine, and lysine by decarboxylation [25].

Glutamic acid, glycine, and alanine make important contributions to the characteristic flavor and taste of fish [27]. Concentrations of glycine and alanine in grouper increased to 129.94 mg/100 g and 55.06 mg/100 g after 2 days’ storage at 4 °C, and decreased on day 10. Glutamic acid decreased from 17.95 mg/100 g on day 0 to 5.53 mg/100 g on day 2, and then fluctuated between 16.04 and 21.63 mg/100 g. The decrease of glutamic acid may be alanine derived from pyruvate by the transamination of glutamic acid [25].

### 2.3. Total Viable Count and Bacterial Distribution

The bacteria sampled from grouper fillets were counted and isolated on different media. First, a total number of bacteria was counted on plate count agar (PCA). When aerobic plate count (APC) reaches 7–8 log10 (CFU/g), it is considered to have reached its sell-by date [2,28]. The shelf life of grouper was 6–8 days at 4 °C, and the number of bacteria reached 8.3 log10 (CFU/g) on day 8. As shown in Figure 1, the number of psychrotrophs was 1.5 log10 (CFU/g) on day 0, which was lower than that of mesophiles (2.6 log10 (CFU/g)), but these numbers increased after 8 days to 8.4 and 8.1 log10 (CFU/g), respectively. At the beginning of storage, the catabolism of protein in fish provided rich nutrients for the growth of microorganisms, and the total number of all microorganisms increased rapidly. Afterward, the low temperature made the psychrotrophs more competitive than the mesophiles, and the growth of the latter was suppressed. The difference in bacterial counts between the two at the end of storage was reduced, indicating that the psychrotrophs became the dominant bacteria after a certain period of refrigeration. After 16S rRNA gene amplification and sequencing, about 86.67% of all strains were identified as *Pseudomonas* on the CFC plates. This is due to the high selectivity of the culture medium and the universal distribution of *Pseudomonas* on aquatic products. However, on other plates, the *Pseudomonas* count was still high, accounting for 16.67% on MRS, 40.00% on Iron Ager (IA), and 57.10% on violet red bile dextrose agar (VRBGA), respectively. Except for *Pseudomonas*, *Macrococcus*, *Staphylococcus*, and *Psychrobacter* were identified on MRS, and *Shewanella* and *Kocuria* were identified on VRBGA, respectively. Clearly, it could be seen that *Pseudomonas* was the main flora of grouper under 4 °C storage conditions. This was not surprising, because Casaburi [9] and Huang [29] concluded that despite the large number of microorganisms that make up the initial microbiota, only a few species were sufficient to cause spoilage. This result is consistent with the next-generation sequencing (NGS) result.

The other biological factor considered to accelerate spoilage is H_2_S-producing bacteria [2]. In this study, H_2_S-producing bacteria cultured from IA increased from 2.1 log10 (CFU/g) on day 0 to 6.7 log10 (CFU/g) on day 8 and remained stable. For example, *Shewanella putrefaciens* was isolated from IA, which is the main H_2_S-producing bacteria in most seafood [30,31,32]. 

In our work, H_2_S-producing bacteria and *Pseudomonas* were the main microorganisms that caused grouper spoilage during low temperature storage.

### 2.4. Diversity of Microbial Community 

NGS is a useful method for researching microbial communities independent from cultured methods [33]. Sequencing on the Illumina MiSeq platform showed that the average length of these sequences from samples was 423 bp. The alpha diversity estimation and phylotype coverage of bacterial communities in hybrid grouper fillets indicated sufficient sampling readings (Table 3). The coverage of all samples was above 0.999, indicating that the bacteria in all grouper samples could be identified at the current level of sequencing. From the changes of Shannon and Simpson, the microbial diversity was the highest in fish when they were received, and decreased continuously until the end of the storage. At the same time, the values of ACE and Chao were the smallest on day 10, indicating that the community richness was the lowest at the end of storage.

### 2.5. Bacterial Community Diversity and Succession in Grouper Fillets at Family and Genus Levels

Under storage conditions, the overall microbial community diversity in hybrid grouper is affected by storage temperature [34]. On day 0, 35 genera were observed in the grouper samples, but the number decreased to 19 on day 10 (Figure 2). During storage, there were 12 and 3 newly-observed genera on days 6 and 10, respectively. This phenomenon may indicate that some of the genera were only able to grow under an oxygen-rich environment, or that some required amino acids degraded by proteins. In this study, huge changes in the microbial community were observed in the grouper fillets stored at 4 °C for 10 days. Although the original indigenous microbiota was diverse, a small proportion of taxa dominated. The dominant taxa underwent tremendous changes over time, indicating that storage conditions had an important effect on fish. The diversity of the dominant taxa decreased with increasing storage time. Similarly, the gradual decrease in the number the genera from high to extremely low indicated the succession of microbial communities.

### 2.6. Microbiological Analysis and Amplicon Sequencing

This study sought to describe the effect of storage time on the microbial community in groupers to explore the spoilage mechanisms of fish. It was observed that there were huge changes in the microbiota at the phylum, genus, and species levels over the 10 days. 

A total of 183 different OTUs were obtained by 16S rRNA amplicon sequencing. At the beginning of storage, *Macrococcus spp.* was the main genus in the bacterial communities of grouper (Figure 3a). However, the number dropped rapidly. On the other hand, *Pseudomonas spp.* grew at an extremely high speed and became the dominant genus over the course of the storage (increasing from 1.91% on day 0 to 74.38% on day 6) The other two genera that underwent obvious increases were *Psychrobacter* and *Vibrio*, which increased from 0.03% and 0.77% to 8.21% and 13.85%, respectively. Compared to the initial status, the bacterial communities were less abundant. The main microbiome was *Pseudomonas* (97.54%) at the end of the testing period. This was consistent with the identification results obtained by 16s RNA gene amplification sequencing. It was found that 33.58% of *Pseudomonas* were unknown species, which was concordant with the research on meat and Arctic fishes [35]. It can be indicated that *Pseudomonas* has a high degree of heterogeneity and biodiversity in its species. The other 64.85% were identified to be *Pseudomonas azotoformans*, which can be considered as the SSO of hybrid grouper under 4 °C storage conditions. *Pseudomonas* was previously found to be the major-specific spoilage organism in aerobically-stored fish [7], and it has been reported as the most common spoilage bacterium to cause rapid and intense spoilage [7,36]. 

It was noteworthy that the proportion of *Shewanella* was relatively low (Figure 3b). Nevertheless, the result of our study was inconsistent with other research. The reason for this phenomenon may be the DNA extraction and PCR amplification that affected the obtained bacterial community profiles; with fewer PCR cycles, less sample information can be captured [35,37].

### 2.7. Heatmap Analysis

The established hierarchical clustering heatmap at the genus level was based on the top 30 most abundant genera to analyze and compare the composition and dynamic changes of microbial communities in samples at different storage times (Figure 4). The gene tree to the left of the heatmap displays the clustering of genes. The heatmap reflected the changes in some low-abundance microbiota which were not well represented in their histograms (Figure 4) [38].

According to the heatmap, the dominant bacterial community in fresh grouper was *Macrococcus* (87.42%), including 3.38% *Staphylococcus*, 2.05% *Bacillus*, and 2.04% *Acinetobacter*. However, the number of these three bacteria and some other genera decreased and even dropped to 0, suggesting that they were less competitive than *Pseudomonas* during cold storage. *Pseudomonas* has been confirmed as SSO in several refrigerated fish [39,40,41], and it grows rapidly during refrigeration and plays an important role in the microbial community [41], as supported by the results of this study.

### 2.8. VOCs Analysis on Grouper Fillets

The process of microbial spoilage results in the sensory deterioration of grouper fillets during storage; therefore, it is suggested that the growth of microbial is closely related to sensory evaluation at the end of shelf life. The levels of the 28 VOCs in fish stored at 4 °C are given in Table 4. It was found that volatile compounds, which were extracted and detected by HS-SPME-GC-MS, changed during storage. Most VOCs were metabolites of microbial origin, such as alcohols and aldehydes.

At the beginning of the experiment, 1-penten-3-ol, and 1-octen-3-ol increased from 2.43 (day 0) to 3.91 (day 4) and 18.00 (day 0) to 33.47 (day 4), respectively. The opposite pattern was observed with 2-ethyl-1-hexanol, which decreased from 207.39 (day 0) to 137.07 (day 4). A similar finding has been reported in other studies [10,42]. Some scholars pointed out that the effect of 15-lipoxygenase on eicosapentaenoic acid (EPA) and 12-lipoxygenase on arachidonic acid can explain the increase in 1-penten-3-ol and 1-octen-3-ol [43,44]. At the middle and later stages of the experiment, the production of alcohol increased significantly. After 10 days of storage, the relative level of 1-octen-3-ol increased significantly to 40.85 (day 10) (*p* < 0.05). This result was consistent with the findings of Josephson and Lindsay [45] on freshly-harvested fish which had low levels of 1-octen-3-ol (10–100 ppb) at the beginning of the storage. Additionally, 1-octen-3-ol has a low odor detection threshold, and its production is mainly due to the spoilage of fish: therefore, 1-octen-3-ol is a good quality indicator in fish [46].

At the early stage of storage, the relative content of most carbonyls changed significantly [8]. The VOCs of the fresh grouper were mainly comprised of aldehydes such as hexanal, heptanal, octanal, nonanal, and decanal. Among these chemicals, straight-chain aldehydes are characterized as having a green, herbal odor [47]. However, when their concentration exceeds a certain threshold, they become undesirable [48]. For example, when the concentration of hexanal exceeds a certain threshold, it is related to rancid odors in stored meat [45]. As reported by Marsili [49,50], hexanal is the predominant autoxidation byproduct of linoleic and linolenic acid. There are many ketones in fish because of the presence of bacteria [9]; however, in this study, only a small number of ketones were detected. This phenomenon occurred when the temperature exceeded 100 degrees [51]. The ketones were identified as 2-nonanone and acetylacetone over the course of the experiment.

It was reported by Casaburi [9] that *Pseudomonas* and *Carnobacterium* are the main bacteria that produce alcohols and aldehydes. The number of *Pseudomonas* and *Carnobacterium* increased with storage time in this study. However, the diversity of microorganisms during storage increased the complexity of analyzing volatile substances; further studies are needed on the effects of different microorganisms on odor changes.

### 2.9. VOCs Analysis of Bacteria

The undesirable odors generated during spoilage are related to bacterial metabolism [52]. A total of 154 VOCs were detected in 20 samples and were classified into 11 categories (Figure 5). The number of aromatic compounds for almost all bacteria strains was identified as 15~25 types.

In the heatmap (Figure 5), 20 groups of bacteria and 11 kinds of aromatic compounds were correlated. As shown in the figure, the group of *Pseudomonas rhodesiae*, *Pseudomonas antarctica*, *Pseudomonas azotoformans*, *Pseudomonas koreensis*, *Pseudomonas moraviensis*, and *Pseudomonas extremaustralis* were highly related to the production of pyrazines and hydrocarbons, while *Macrococcus caseolyticus*, *Moraxella osloensis*, *Shewanella putrefaciens*, *Staphylococcus saprophyticus*, *Pseudomonas putida*, and *Pseudomonas helmanticensis* were less related to these aromatic compounds. but produced higher levels of alcohol and other compounds. It was observed that a high content of alcohol was produced by *Citrobacter murliniae*, *Staphylococcus edaphicus*, *Bacillus velezensis*, and *Psychrobacter faecalis*. As for hydrocarbons, they were concentrated in the genus of *Pseudomonas*, especially on *Pse. rhodesiae*. With respect to ethers, they were the most abundant category, and were not detected in other bacteria apart from the group of *Pse. Songnenensis*, *Pseudomonas psychrophila*, *Pseudomonas parafulva*, and *Klebsiella oxytocahigh*. In particular, *Pseudomonas* was considered to be a great contributor to ester production and to play an important role in sensory evaluation [53,54].

Microorganisms utilize different precursor compounds to produce their volatile metabolites that significantly alter the composition of their VOCs [55]. Many researchers [37,54,56] have stated that the existence of 2-ethyl-1-hexanol was associated with *Pseudomonas* activity. To a certain extent, *Pseudomonas* also participated in the production of 3-methyl-1-butanol, 2-heptanone, 2-nonanone, and 2-undecanone [5,52,57], which was basically consistent with our findings.

## 3. Materials and Methods

### 3.1. Grouper Fillet Provision and Storage

Ten hybrid groupers (length: 54.60 ± 1.78 cm, weight: 1100 ± 55 g) were purchased from an aquatic product market of Luchaogang, Shanghai, China. The fish were then placed in polyethylene bags with oxygen and water and transported to the laboratory alive at 0.5 h. The hybrid groupers were stunned, headed, gutted, and washed using deionized water. The fish were cut into 8 slices (25 ± 5 g) using sterile cutting boards and knives. All the slices were then mixed, and a random sample of 10 slices was placed in one sterile plastic bag (240 ± 20 g). Eight bags of samples were stored at 4 ± 1 °C until required. One bag was selected for all analyses at each specific storage time. Sampling times were days 0, 2, 4, 6, 8, and 10.

### 3.2. Physico-Chemical Analysis

The method described by Benjakul [58] was slightly modified. The color changes of hybrid grouper were determined by the colorimeter (Chroma Meter CR-400, Hangzhou Ke Sheng Instrument Co., Ltd. China), and the weight was calculated according to the following formula:(1)$$W^{*}\left(\mathrm{wight}\right)=100-\sqrt{{\left(100-L^{*}\right)}^{2}+b^{*2}+a^{*2}}$$

The TVB-N was Determined According to the Method Fan & Luo [59], and Expressed in mg/100 g of Fish. The Data were Analyzed in Triplicate Using Excel 2019.

### 3.3. FAAs Analysis

A 2 g sample was homogenized in 10 mL 5% trichloroacetic acid solution using an Ultra Turrax T25 (IKA, Staufenim, Germany). After that, the samples were centrifuged at 10,000 g for 10 min at 4 °C. The extraction and centrifugation were repeated twice, and the supernatant after centrifugation was diluted to 25 mL. One milliliter of the extract was filtered using a 0.22 μm membrane filter, and then the filtrate was used for FAA quantification using an automatic amino acid analyzer (Agilent 1100 Series; Palo Alto, CA, USA). The data were analyzed in triplicate.

### 3.4. Determination of VOCs in Grouper Fillets

The method described by Iglesias [10] has been slightly modified. At every sampling point, 5 g of sample fillets were stored at −40 °C and analyzed within 3 weeks. Each 2.5 g portion was transfered into a 15 mL glass vial with 2.5 mL of saturated NaCl solution, and stirred 5 min. To remove any volatile contaminants, the fiber was exposed to the inlet for 10 min before each analysis. The 65 μm SPME fiber (PDMS/DVB) was then exposed to the headspace under the same conditions for 20 min at 40 °C.

An Agilent 7890A gas chromatograph (Agilent Technologies Inc., Santa Clara, California, USA) was used with an Agilent 5975C mass spectrometer to analyze VOCs. The constant flow rate of helium was 0.9 mL/min as a carrier gas. The temperature of the oven was held for 5 min at 40 °C, and then raised to 150 °C at a rate of 4 °C/min; subsequently, it was raised to 250 °C at a rate of 30 °C/min and maintained for 5 min. The interface temperature was set at 280 °C. VOCs were identified via the comparison of mass spectra with existing data in the NIST11 library. The content changes of each compound were calculated based on the relative peak areas (peak areas/10^6^). The data were analyzed in triplicate.

### 3.5. Microbial Diversity Analysis

Microbial DNA was extracted from four groups of samples (day 0, day 6, day 10) using the E.Z.N.A. soil DNA Kit (Omega Bio-Tek, Norcross, GA, U.S.). The DNA quality was checked by 1% agarose gel electrophoresis. The diluted-genomic DNA was used as a template, and the V3–V4 hypervariable regions of the bacteria 16S rRNA gene were amplified with primers by the thermocycler PCR system (Thermo Fisher Scientific, Waltham, Massachusetts, USA). Each sample was subjected to three replicate experiments. The reference point was subjected to preliminary quantification and the PCR product was quantified by QuantiFlour TM-ST Blue Fluorescence System (Promega) and then mixed according to the sequencing amount of each sample. According to the standard procedure of Majorbio Bio-Pharm Technology Co. Ltd. (Shanghai, China), the combined equimolar purified amplicons were paired-end sequenced on an Illumina MiSeq platform.

### 3.6. Microbiological Analysis

#### 3.6.1. Enumeration and Purification of Bacteria

Twenty-five grams of samples from each hybrid grouper were aseptically weighed and homogenized in 225 mL sterilized 0.85% sterile NaCl solution. Ten-fold serial dilutions were performed. Five media were selected for the detachment and counting of residential bacteria. Total viable counts (TVC) and psychrotrophs were determined on PCA, respectively, incubated at 35 ± 1 °C for 48 h and incubated at 4 ± 1 °C for 10 days. H_2_S-producing bacteria were enumerated on IA at 20 ± 1 °C for 4 days. Lactic acid bacteria (LAB) were enumerated at 30 ± 1 °C for 48 h using MRS agar. The Enterobacteriaceae were enumerated on VRBDA and incubated at 35 ± 1 °C for 48 h. *Pseudomonas sp.* were enumerated on *Pseudomonas* CFC Selective Agar at 20 ± 1 °C for 72 h. All of these media were procured from HaiBo Biological Technology Co., Ltd. (Qingdao, China). Bacterial concentration was expressed as log10 (CFU/g) [60]. Colonies were randomly selected from the incubated plates and streaked on PCA repeatedly to obtain pure colonies [61]. Then, individual strains were preserved in tryptic soy broth (TSB) with 30% glycerol at −80 °C.

#### 3.6.2. Isolation and Identification of Spoilage Bacteria

Colonies with larger decomposition zone were selected from each selective medium and strewed twice on the medium to obtain individual colony. The universal primers 27F (5′-AGAGTTTGATCCTGGCTCAG-3′) and 1492R (5′-CGACGGCTACCTTGTTACGA-3′) were used to amplify the 16S rRNA gene sequences for identification. The PCR products were purified and sequenced by Personalbio (Shanghai Personal Gene Technology Co., Ltd., Shanghai, China).

#### 3.6.3. HS-SPME-GC-MS analysis

Minor modifications were made to the method described in Yuxiang, Z. [53]. Twenty-two representative bacteria were selected and incubated in TSB at 4 °C for 10 days, and then centrifuged 5 mL of the bacterial solution. To promote the precipitation of volatiles, the supernatant was placed in a 15 mL glass vial with 1.5 g NaCl.

The volatiles were extracted by equilibrating with a 50/30 μm DVB/CAR/PDMS fiber (Supelco, Bellefonte, PA, USA) at 40 °C for 15 min. VOC generation was measured using a GC–MS 7890A-5975C Ultra system (Agilent Technologies Inc., USA). High purity helium was still used as the carrier gas at a column flow of 0.9 mL/min. The temperature of the oven held for 3 min at 40 °C, and then increased to 120 °C at a rate of 4 °C/min; subsequently, it increased to 250 °C at a rate of 6 °C/min and was maintained for 10 min. The interface temperature was set at 230 °C. VOCs were identified via the comparison of mass spectra with existing data in the NIST11 library. The content changes of each compound were calculated based on the relative peak areas (peak areas/10^6^).

### 3.7. Statistical Analysis

All physico-chemical determinations were done in triplicate. Differences in color, TVB-N, FAAs, and bacterial counts were statistically tested by performing Analysis of Variance. SPSS 22.0 (SPSS Inc., Chicago, IL, USA) was used for a one-way analysis of variance. Frequency percentage (FP) was used to analyze the distribution of dominant bacteria in fish samples at different storage times. In GC-MS analysis, the data were first standardized, and then analyzed using the R software to generate the heatmap.

## 4. Conclusions

Changes in the microbial community and physico-chemical quality of hybrid grouper preserved at 4 °C for 10 days were described in this paper. The values of whiteness and TVB-N increased with storage time. Glutamate, glycine, and alanine, as the main contributors to flavor, changed significantly (*p* < 0.05) over time. The number of bacteria in grouper reached 6.8 log10 (CFU/g) on day 6, which indicated that the sample was spoiled. The VOCs during the storage of grouper filltes were dissected, and a large number of alcohols, aldehydes, and ketones were generated due to microbial action. Both 1-penten-3-ol and 1-octen-3-ol were identified as potential CSIs candidate for grouper. The detection of VOCs produced by microorganisms in separate cultures showed that the production of 2-ethyl-1-hexanol, 3-methyl-1-butanol, 2-heptanone, 2-nonanone, and 2-undecanone was related to the growth and reproduction of *Pseudomonas ssp.* to some extent. The results of the study provide valuable information for the quality control of grouper freshness.

In order to confirm and understand the mechanism of production of volatile organic compounds, further studies on VOCs produced by inoculating individual SSOs in sterile fish are needed.

## Figures and Tables

**Figure 1 molecules-25-00818-f001:**
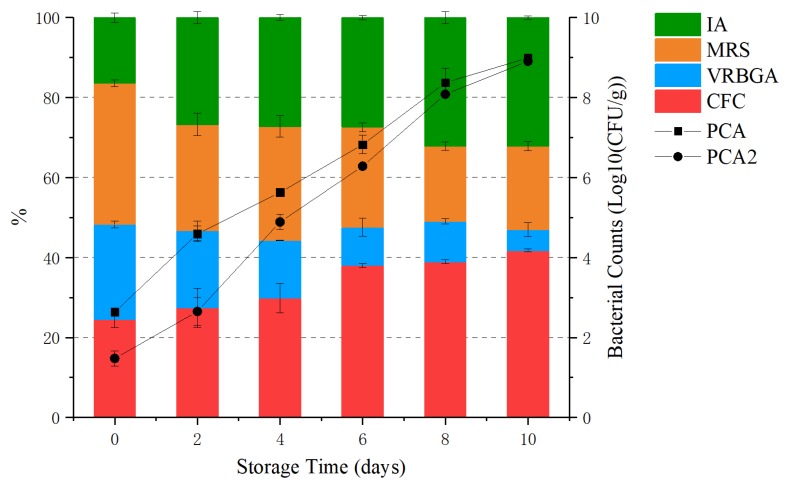
Total viable count of bacteria on five types of media. PCA = Plate Count agar; CFC = *Pseudomonas* CFC Selective agar; VRBGA = violet red bile dextrose agar; MRS = de Mann, Rogosa, Sharpe agar; IA = Iron agar, PCA2 = PCA incubated for psychrotrophs.

**Figure 2 molecules-25-00818-f002:**
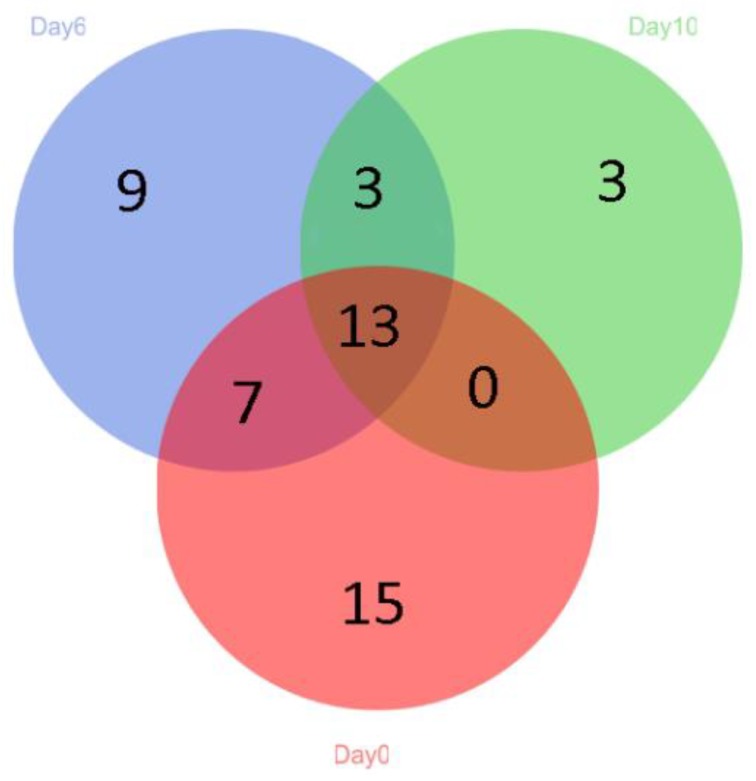
Venn diagram showing the distribution (97% similarity) of the genera levels present in the samples of grouper stored at 4 °C for 10 days. Each number represents a shared or unique genus in the samples in each sampling day. Different color indicates different sampling days (day 0 = red, day 6 = blue and day 10 = green).

**Figure 3 molecules-25-00818-f003:**
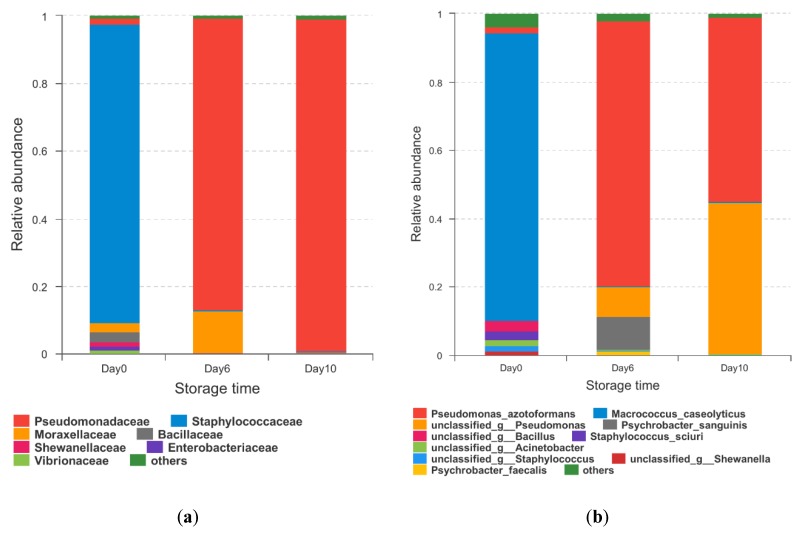
Community barplot analysis of microbiota at the genus level (**a**) and species level (**b**) from grouper fillets during storage.

**Figure 4 molecules-25-00818-f004:**
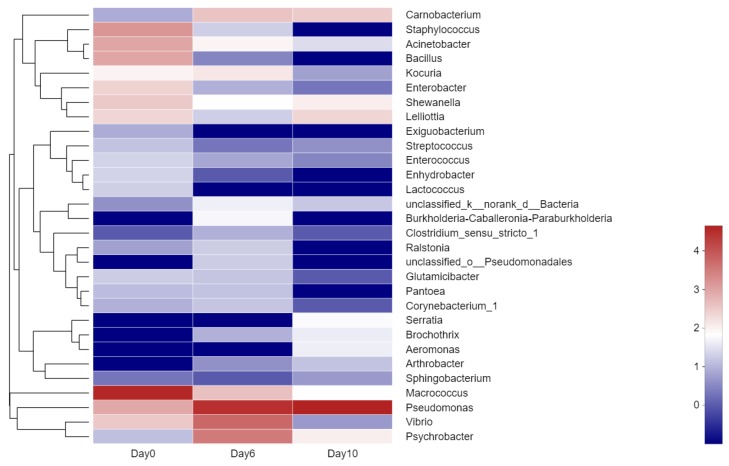
Community heatmap at the genus level of microbiota from hybrid grouper during storage at 4 °C. Those with the highest abundance relationships are in red, while those with the lowest abundant relationships are blue.

**Figure 5 molecules-25-00818-f005:**
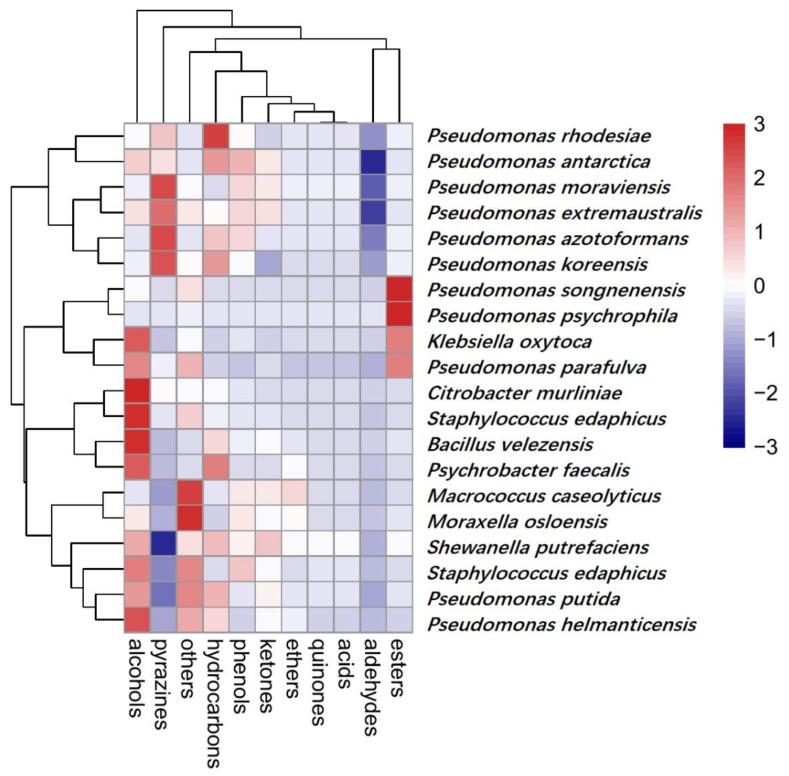
Heatmap showing the relative content of various aroma categories in isolates. Those with the highest abundant relationships are in red, while those with the lowest abundant relationships are in blue.

**Table 1 molecules-25-00818-t001:** Changes in the color and TVB-N of hybrid grouper fillets during storage at 4 °C.

Time (days)	*L* *	*a* *	*b* *	W *	TVB-N (mg/100 g)
0	41.55 ± 1.39 ^c^	−0.46 ± 0.14^bc^	−2.78 ± 1.50 ^bc^	41.46 ± 1.37 ^c^	9.60 ± 0.41 ^e^
2	43.52 ± 0.34 ^bc^	0.53 ± 0.09^a^	−3.63 ± 0.26 ^c^	43.40 ± 0.33 ^bc^	10.27 ± 0.27 ^e^
4	44.21 ± 0.91 ^b^	−0.28 ± 0.37^bc^	−4.01 ± 0.18 ^c^	44.06 ± 0.91 ^b^	12.53 ± 0.86 ^d^
6	43.36 ± 1.20 ^bc^	−0.86 ± 0.21^c^	−3.30 ± 1.09 ^bc^	43.25 ± 1.26 ^bc^	14.26 ± 1.08 ^c^
8	46.79 ± 1.90 ^a^	0.89 ± 0.86^a^	3.86 ± 0.55 ^a^	46.63 ± 1.94 ^a^	20.21 ± 0.99 ^b^
10	47.49 ± 0.76 ^a^	0.16 ± 0.36^ab^	−1.87 ± 0.60 ^b^	47.45 ± 0.77 ^a^	21.36 ± 2.07 ^a^

**W *** = weight of hybrid grouper. The different superscripts (a ~ e) in the column denote significant differences (*p* < 0.05). The same as below.

**Table 2 molecules-25-00818-t002:** Changes in the FAA contents (mg/100g) of hybrid grouper fillets during storage at 4 °C.

Amino Acids	Day 0	Day 2	Day 4	Day 6	Day 8	Day 10
Aspartic acid	1.50 ± 0.04 ^d^	1.94 ± 0.08 ^c^	1.05 ± 0.09 ^e^	3.04 ± 0.02 ^b^	1.66 ± 0.00 ^d^	3.48 ± 0.02 ^a^
Threonine	13.31 ± 1.42 ^d^	21.89 ± 0.15 ^a^	19.95 ± 0.96 ^ab^	16.99 ± 1.32 ^bc^	17.37 ± 0.72 ^bc^	14.39 ± 0.04 ^cd^
Serine	6.20 ± 0.64 ^b^	16.02 ± 0.03 ^a^	16.60 ± 0.87 ^a^	4.73 ± 3.68 ^b^	8.39 ± 0.36 ^b^	6.33 ± 0.01 ^b^
Glutamic acid	17.95 ± 1.76a ^bc^	5.53 ± 0.03 ^d^	20.30 ± 0.60 ^ab^	17.61 ± 1.80 ^bc^	21.63 ± 0.63 ^a^	16.04 ± 0.39 ^c^
Glycine	69.69 ± 7.22 ^cd^	129.94 ± 1.19 ^a^	95.6 ± 2.75 ^b^	86.88 ± 6.35 ^b^	85.25 ± 4.36 ^bc^	58.37 ± 1.01 ^d^
Alanine	30.14 ± 3.46 ^c^	55.06 ± 0.09 ^a^	42.06 ± 1.93 ^b^	45.19 ± 3.56 ^b^	33.73 ± 1.70 ^c^	33.07 ± 0.48 ^c^
Valine	4.30 ± 0.40 ^b^	2.32 ± 0.02 ^c^	3.97 ± 0.20 ^b^	4.36 ± 0.28 ^b^	4.72 ± 0.25 ^ab^	5.43 ± 0.09 ^a^
Methionine	2.37 ± 0.20 ^ab^	0.74 ± 0.01 ^d^	1.66 ± 0.09 ^c^	2.05 ± 0.13 ^bc^	2.15 ± 0.15 ^b^	2.61 ± 0.03 ^a^
Isoleucine	2.90 ± 0.20 ^b^	1.56 ± 0.01 ^d^	2.08 ± 0.09 ^c^	2.99 ± 0.17 ^b^	3.07 ± 0.18 ^b^	3.66 ± 0.05 ^a^
Leucine	4.44 ± 0.33 ^b^	2.56 ± 0.01 ^d^	3.49 ± 0.13 ^c^	4.83 ± 0.25 ^b^	5.09 ± 0.28 ^b^	6.06 ± 0.11 ^a^
Tyrosine	1.86 ± 0.09 ^b^	1.03 ± 0.01 ^d^	1.50 ± 0.07 ^c^	1.88 ± 0.08 ^b^	1.98 ± 0.16 ^b^	2.37 ± 0.06 ^a^
Phenylalanine	2.07 ± 0.12 ^c^	0.96 ± 0.07 ^d^	1.34 ± 0.08 ^c^	1.73 ± 0.09 ^b^	1.71 ± 0.05 ^b^	1.88 ± 0.01 ^bc^
Lysine	30.74 ± 2.84 ^b^	21.06 ± 0.00 ^c^	20.57 ± 0.93 ^c^	27.46 ± 1.52 ^b^	31.02 ± 1.15 ^b^	36.44 ± 0.16 ^a^
Histidine	2.86 ± 0.3 ^b^	2.65 ± 0.06 ^b^	3.88 ± 0.26 ^a^	3.88 ± 0.31 ^a^	3.95 ± 0.12 ^a^	3.19 ± 0.05 ^ab^
Arginine	5.91 ± 0.53 ^bc^	3.68 ± 0.00 ^e^	5.45 ± 0.09 ^cd^	6.77 ± 0.40 ^b^	4.48 ± 0.20 ^de^	8.24 ± 0.02 ^a^
Total	196.24 ± 27.66 ^c^	266.93 ± 2.11 ^a^	239.48 ± 12.88 ^ab^	230.4 ± 17.75 ^bc^	226.21 ± 14.56 ^bc^	201.56 ± 2.72 ^c^

The different superscripts (a ~ e) in the column denote significant differences (*p* < 0.05).

**Table 3 molecules-25-00818-t003:** Phylotype coverage and alpha diversity estimation of bacterial communities in hybrid grouper fillets during storage at 4 °C.

Sample	OTUs	Shannon	Simpson	ACE	Chao	Coverage
Day 0	53	1.359056	0.388594	105.7616	138.25	0.9997
Day 6	51	0.929429	0.614348	141.148483	146	0.9996
Day 10	39	0.845352	0.621279	50.48075	46.58333	0.9999

OTUs: operational taxonomic units; Shannon: the Shannon index of community diversity; Simpson: the Simpson index of community diversity; ACE: the ACE estimator; Chao: the Chao estimator; Coverage: the good′s community coverage.

**Table 4 molecules-25-00818-t004:** Volatile compounds of hybrid grouper by HS-SPME-GC-MS at 4 °C.

VOCs	Day 0	Day 2	Day 4	Day 6	Day 8	Day 10
*Alcohols*						
1-Penten-3-ol	2.43 ± 0.01 ^c^	2.40 ± 0.05 ^c^	3.91 ± 0.16 ^d^	ND	3.12 ± 0.1 ^b^	3.84 ± 0.83 ^a^
1-Heptanol	ND	ND	13.11 ± 2.67 ^a^	8.71 ± 4.33 ^b^	ND	ND
1-Hexanol	ND	ND	ND	ND	7.71 ± 4.24 ^a^	ND
1-Octen-3-ol	18.00 ± 2.83 ^c^	21.92 ± 14.21 ^bc^	33.47 ± 9.47 ^ab^	39.19 ± 4.00 ^a^	16.59 ± 4.64 ^c^	40.85 ± 3.61 ^a^
2-Ethyl-1-hexanol	207.39 ± 17.32 ^c^	127.28 ± 2.00 ^d^	137.07 ± 8.22 ^d^	568 ± 0.01 ^a^	227.46 ± 8.87 ^c^	421.15 ± 19.95 ^b^
1-Pentanol	ND	ND	ND	5.72 ± 5.00 ^a^	ND	ND
*Aldehydes*						
Hexanal	51.97 ± 7.07 ^d^	84.76 ± 11.82 ^bc^	111.25 ± 7.91 ^a^	64.91 ± 4.7 ^c^	34.91 ± 3.27 ^e^	ND
Heptanal	15.03 ± 3.99 ^bc^	16.43 ± 2.00 ^bc^	21.58 ± 2.59 ^a^	16.64 ± 2.15 ^b^	10.70 ± 3.61 ^c^	3.78 ± 1.04 ^d^
Benzaldehyde	16.39 ± 2.6 ^ab^	13.53 ± 0.95 ^b^	21.97 ± 7.10 ^a^	13.31 ± 4.03 ^b^	ND	ND
(E)-2-Octenal	ND	ND	6.78 ± 0.00 ^a^	ND	ND	ND
Nonanal	152.70 ± 11.36 ^a^	31.20 ± 8.51 ^cd^	55.33 ± 1.05 ^b^	41.89 ± 5.99 ^c^	18.32 ± 3.95 ^e^	20.76 ± 1.07 ^de^
Decanal	4.60 ± 1.41 ^a^	ND	3.80 ± 0.94 ^a^	ND	ND	3.27 ± 0.33 ^a^
*Ketones*						
Acetylacetone	ND	ND	ND	ND	ND	1.11 ± 0.99 ^a^
2-Nonanone	ND	ND	3.63 ± 0.58 ^b^	2.79 ± 0.10 ^c^	ND	6.15 ± 0.99 ^a^
*Others*						
Trimethylamine	ND	ND	ND	ND	0.40 ± 0.52^a^	ND
*Hydrocarbon*						
Nonane	6.34 ± 0.98 ^bc^	ND	ND	ND	ND	8.43 ± 0.95 ^a^
Decane	ND	ND	ND	ND	ND	19.26 ± 1.34 ^a^
Undecane	9.11 ± 1.78 ^a^	ND	7.61 ± 0.01 ^b^	ND	6.33 ± 0.80 ^b^	ND
Naphthalene	4.90 ± 0.1 ^a^	ND	6.59 ± 0.92 ^a^	ND	4.60 ± 3.61 ^a^	5.50 ± 2.52 ^a^
Dodecane	28.07 ± 6.04 ^a^	8.46 ± 2.18 ^c^	17.09 ± 2.65 ^b^	6.51 ± 0.87 ^c^	7.96 ± 1.00 ^c^	4.23 ±0.10 ^c^
Tridecane	53.45 ± 0.31 ^a^	11.94 ± 0.83 ^c^	30.09 ± 7.51 ^b^	9.19 ±1.00 ^c^	13.08 ± 1.89 ^c^	8.61 ± 0.52 ^c^
Tetradecane	15.24 ± 5.29 ^ab^	ND	11.63 ± 0.00 ^bc^	15.27 ± 1.99 ^ab^	10.47 ± 1.00 ^c^	18.63 ± 1.42 ^a^
Pentadecane	328.65 ± 24.97 ^a^	37.98 ± 2.04 ^b^	312.99 ± 60.00 ^a^	338.00 ± 18.22 ^a^	103.08 ± 7.92 ^b^	365.61 ± 60.00 ^a^
Hexadecane	7.95 ± 0.00 ^a^	ND	6.23 ± 0.00 ^a^	ND	ND	9.09 ± 0.00 ^a^
Heptadecane	54.73 ± 4.58 ^a^	36.32 ± 3.82 ^c^	45.26 ± 4.30 ^b^	42.83 ± 6.99 ^bc^	6.61 ± 1.92 ^d^	54.25 ± 0.00 ^a^
Pentadecane	111.4 ± 2.56 ^a^	46.73 ± 1.92 ^d^	84.22 ± 0.21 ^c^	99.72 ± 7.01 ^b^	23.66 ± 0.34 ^e^	ND
Styrene	5.14 ± 0.86 ^bc^	ND	ND	3.9 ± 1.20 ^c^	5.91 ± 0.57 ^b^	11.17 ± 2.02 ^a^
Caryophyllene	18.86 ± 0.01 ^a^	ND	13.69 ± 0.24 ^b^	17.49 ± 1.06 ^ab^	ND	21.97 ± 5.94 ^a^

Values are means of three independent determinations (*n* = 3); ND, not detected. Values in the same row followed by different superscript letters are significantly different (*p* < 0.05).

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
