# Peer review of "Characterization of the Volatiles and Quality of Hybrid Grouper and Their Relationship to Changes of Microbial Community During Storage at 4 °C"

_molecules, 2020, doi:10.3390/molecules25040818_

Round 1

Reviewer 1 Report

The research article represents valuable scientific findings on the microbial community and quality attributes of hybrid grouper at the post-mortem period during cold storage. Please check out my comments below.

The H of “Hybrid” should be lowercase in the research title. Please be consistent when using the units “mL” (line 285, line 299 and line 330) throughout the manuscript. In order to be consistent with the formality in the context (line 312-316) with the usage of “±”, there should be a space before and after “±” in Table 1, Table 2 and Table 4. The resolution of Figure 4 is poor and the authors should improve it.

Author Response

Dear reviewer,

We really appreciate your suggestions to improve the manuscript. We revised the manuscript carefully and highlighted the changes in the text with red color. Enclosed, please find our summarized responses to your comments/suggestions and the revised manuscript for your review and/or final approval for publication.

Point 1: The H of “Hybrid” should be lowercase in the research title. Please be consistent when using the units “mL” (line 285, line 299 and line 330) throughout the manuscript. In order to be consistent with the formality in the context (line 312-316) with the usage of “±”, there should be a space before and after “±” in Table 1, Table 2 and Table 4. The resolution of Figure 4 is poor and the authors should improve it.

Response 1: We changed what the reviewer suggested. And we carefully read, checked and corrected errors we could find.

Point 2: The resolution of Figure 4 is poor and the authors should improve it.

Response 2: We replaced Figure 4 with a higher resolution picture.

Reviewer 2 Report

The manuscript “Characterization of the volatiles and quality of Hybrid grouper and their relationship to changes of microbial community during storage at 4 ºC” is well organized but the English must be improved in some specific parts. The several applied methods were appropriate to try to correlate changes in VOCs and the microbial community present in grouper fillets during cold storage. However, specific details about the experimental conditions must be included to better understand and discuss the obtained results. Other improvements and corrections must be implemented, to know:

Keywords: replace “microbiota composition” by “microbiota diversity”; replace specific spoilage organism by for example concrete genus of microorganisms.

There are several abbreviations during the manuscript please include the full description when they appear the first time. In my opinion the description in the abstract not counts.

p2. Line 53 – Improve the sentence

p2. line 61-62 – Confuse sentence. Rewrite

Table 1 – identify “W” in the table legend or foot note

p3. line 105-108 – too many ideas in the same sentence. Please rewrite.

p3. line 107 – “the number of psychrotrophs was lower than the number of mesophile obviously, …” – explain why it is obvious.

p3. Line 110 – methodology mixed with results and discussion. Please remove the information about methodology from this section to the methods section.

p4. line 115 – it is important to refer the specific groups of bacteria for the selective media used.

section 2.4 – No comments or discussion about ACE and Chao estimators (table 3) were performed.

p5. line 147 – I think there is a mistake in the number of genus on day 10th.

p6. line 182 – Explain how the DNA extraction and PCR amplification affects the bacterial community

Fig 3 and fig 4 – The letter size of legends must bigger

p6. line 189-191 – include the information in the figure 4 legend

p6. line 195 – You forgot mentioned Staphylococcus

p7 Line 225 – “high concentration of hexanal”. which is concentration level? In Table 5 hexanal reached to 111.25 in day 4. At this level the rancid odors were detected?

p7 Line 229-230 – Explain better

Table 4 – Format the numbers in the table. Include storage days and the “units” of the results.

Figure 5 – species in italic

Section 3.1 – detailed description of the methodology must be performed. For instance: How many fishes were used? How many fillets? fillets dimensions? Weight? How many fillets per bag? How many bags? plastic type of the bags? Sampling for analysis? the same bag for all the analysis at specific storage time, or fillets from different bags?

For analysis indicate the number of replicates.

p11-Line 329 – Information about the culture medium

Section 3.5.3 - Why the method for VOC identification is not the same as in section 3.4

Additional information and specific comments are included in the pdf file attached.

Author Response

Dear reviewer,

We really appreciate your suggestions to improve the manuscript. We revised the manuscript carefully and highlighted the changes in the text with red color. Enclosed, please find our summarized responses to your comments/suggestions and the revised manuscript for your review and/or final approval for publication.

Point 1: Keywords: replace “microbiota composition” by “microbiota diversity”; replace specific spoilage organism by for example concrete genus of microorganisms.

Response 1: We corrected what the reviewer suggested.

Point 2: There are several abbreviations during the manuscript please include the full description when they appear the first time. In my opinion the description in the abstract not counts.

Response 2: We included the full descriptions of those abbreviations when they appear the first time.

Point 3: p2. Line 53 – Improve the sentence

Response 3: We rewrote the sentence (p2. Line 57-58).

Point 4: p2. line 61-62 – Confuse sentence. Rewrite

Response 4: We rewrote the sentence (p2. Line 64-65).

Point 5: Table 1 – identify “W” in the table legend or foot note

Response 5: We added a note of “W” in the foot note of Table 1 (p3. Line 85).

Point 6: p3. line 105-108 – too many ideas in the same sentence. Please rewrite.

Response 6: We rewrote the sentence (p3. Line 113-116).

Point 7: p3. line 107 – “the number of psychrotrophs was lower than the number of mesophile obviously, …” – explain why it is obvious.

Response 7: At the beginning of storage, the community richness was the highest. In the first 2 days, the catabolism of protein in fish provided rich nutrients for the growth of microorganisms, and the total number of all microorganisms will grow rapidly, most of which were mesophile. Afterward, low temperature made the psychrotrophs more competitive than mesophiles, and the growth of mesophile was suppressed (p4. Line 116-119).

Point 8: p3. Line 110 – methodology mixed with results and discussion. Please remove the information about methodology from this section to the methods section.

Response 8: We removed the information about methodology from this section to the methods section.

Point 9: p4. line 115 – it is important to refer the specific groups of bacteria for the selective media used.

Response 9: In this study, 78 strains were identified on 4 selective media by 16S rRNA, however, many of them belongs to the same species. After sifting out congeneric strains, 20 strains were obtained and they were specified in article (p4 Line 128-129).

Point 10: section 2.4 – No comments or discussion about ACE and Chao estimators (table 3) were performed.

Response 10: We added discussions about ACE and Chao estimators (p5. Line 152-153).

Point 11: p5. line 147 – I think there is a mistake in the number of genus on day 10th.

Response 11: Thank you for pointing out my mistake, we corrected “13” to “19”.

Point 12: p6. line 182 – Explain how the DNA extraction and PCR amplification affects the bacterial community

Response 12: The number of PCR cycles can affect sample enrichment. With fewer cycles, less sample information can be captured.

Point 13:  Fig 3 and fig 4 – The letter size of legends must bigger.

Response 13: We replaced the picture of fig 3 and fig 4 with a larger letter in the legend.

Point 14: p6. line 189-191 – include the information in the figure 4 legend

Response 14: We move the information to the legend in Figure 4.

Point 15: p6. line 195 – You forgot mentioned Staphylococcus

Response 15: Thanks for the tip, we added the data of Staphylococcus to the sentence (p6. Line 210).

Point 16: p7 Line 225 – “high concentration of hexanal”. which is concentration level? In Table 5 hexanal reached to 111.25 in day 4. At this level the rancid odors were detected?

Response 16: Data of VOC is relative amount, not an absolute value. It is not accurate to judge the smell of corruption from 111.25 alone. “high concentration of hexanal” means that the concentration of hexanal is related to the rancid odors when it exceeds a certain threshold.

Point 17: p7 Line 229-230 – Explain better

Response 17: We modified the sentence (p8. Line 249-252)

Point 18: Table 4 – Format the numbers in the table. Include storage days and the “units” of the results.

Response 18: We formatted the significant figures for all results in Table 4.

The result was the peak area in the mass spectrum × 10-6, which is a relative quantification and not an absolute value, so the unit is arbitrary [1].

Point 19: Figure 5 – species in italic

Response 19: The pheatmap library (package in R) used for Figure 5 has no option to modify the font, so we edited the italics by photoshop.

Point 20: Section 3.1 – detailed description of the methodology must be performed. For instance: How many fishes were used? How many fillets? fillets dimensions? Weight? How many fillets per bag? How many bags? plastic type of the bags? Sampling for analysis? the same bag for all the analysis at specific storage time, or fillets from different bags?

Response 20: Ten hybrid groupers were stunned, headed, gutted, and washed using deionized water. The fish were cut into 8 slices (25 ± 5 g) by sterile cutting boards and knives. And then mixed all slices, a random sample of 10 slices placed in one sterile plastic bag (240 ± 20 g). Eight bags of samples were stored at 4 ± 1 oC until required. One bag was selected for all analyses at each specific storage time.

Point 21: For analysis indicate the number of replicates.

Response 21: The data were analyzed in triplicate (p10. Line 303-304, p10. Line 311, Line 326).

Point 22: p11-Line 329 – Information about the culture medium

Response 22: The medium used here is TSB (p12. Line 361).

Point 23: Section 3.5.3 - Why the method for VOC identification is not the same as in section 3.4

Response 23: The VOC identification method in Section 3.5.3 was used for bacteria solution, and the method in Section 3.4 was used for fish, so the methods are different.

Iglesias, J.; Medina, I.; Bianchi, F.; Careri, M.; Mangia, A.; Musci, M. Study of the volatile compounds useful for the characterisation of fresh and frozen-thawed cultured gilthead sea bream fish by solid-phase microextraction gas chromatography–mass spectrometry. Food Chemistry 2009, 115, 1473-1478, doi:10.1016/j.foodchem.2009.01.076.
